# “*Do Your Homework as Your Heart Takes over When You Go Looking*”: Factors Associated with Pre-Acquisition Information-Seeking among Prospective UK Dog Owners

**DOI:** 10.3390/ani13061015

**Published:** 2023-03-10

**Authors:** Rebecca Mead, Katrina E. Holland, Rachel A. Casey, Melissa M. Upjohn, Robert M. Christley

**Affiliations:** Dogs Trust, 17 Wakley Street, London EC1V 7RQ, UK

**Keywords:** dogs, dog acquisition, pre-acquisition research, pre-acquisition behaviours, preparatory research

## Abstract

**Simple Summary:**

Dogs are the most common pet animal species in the UK, with many households acquiring dogs every year. However, little is known about whether prospective owners undertake research prior to acquiring a dog and what factors may affect the likelihood of doing so. This paper reports findings from a mixed methods study into dog acquisition in the UK. We found that almost half of existing owners did not look for information or advice before getting their most recently acquired dog, and that those with previous dog ownership experience were less likely to do so than first-time owners. Younger prospective owners were more likely to have undertaken pre-acquisition research, as were those with formal education qualifications. Findings may be of interest to those who provide advice related to dog acquisition and dog ownership, in order to encourage more prospective owners towards effective pre-acquisition research.

**Abstract:**

The factors influencing why and how people decide to acquire a dog are not well understood and little is known about the extent to which prospective owners undertake preparatory research. This study aimed to better understand what factors influence whether prospective dog owners in the UK conduct preparatory research. A 2019 online survey of current (n = 8050) and potential (n = 2884) dog owners collected quantitative and qualitative data. Additional qualitative data were collected through semi-structured interviews with current (n = 166) and potential (n = 10) dog owners. Of the current owners surveyed, 54% stated that they had looked for advice or information prior to acquiring their dog. Of potential owners, 68% reported already having looked for information, while a further 14% were planning to undertake research prior to acquiring a dog. Those with previous dog ownership experience were less likely to undertake pre-acquisition research, as were those who had worked with dogs. Demographic factors were also associated with the likelihood of conducting pre-acquisition research, with younger prospective owners being more likely to have undertaken research, as well as those with formal education qualifications. Among current owners, pre-acquisition research was more likely among those who acquired their dog through a breeder; a specific breed or a mix of two breeds; or as a puppy. Qualitative data were consistent with and added additional understanding and context to these findings. Almost half of current owners did not conduct pre-acquisition research, highlighting the need for increased awareness of its importance and the development of targeted interventions to encourage this activity. Understanding the different factors that influence whether dog owners undertake research may be of interest to animal welfare and veterinary organisations, in order to inform interventions to better prepare people for dog acquisition.

## 1. Introduction

Dogs (*Canis familiaris*) are the most popular companion animal species in the UK today, with an estimated 13 million dogs and 34% of households comprising one or more dogs [1]. There are therefore thousands of prospective owners looking to acquire a new dog each year, and the decisions they make during the acquisition process can have widespread implications for dog welfare. Puppies have been in high demand in the UK, a situation that was particularly notable during the COVID-19 pandemic [2]. Demand has grown for dogs of particularly “fashionable” breeds, including brachycephalic breeds such as French Bulldogs and Pugs [3], despite the health problems faced by these breeds [4,5,6,7,8]. The demand for dogs was such that it was not satisfied by legitimate and reputable UK breeders alone, leading to an increase of unscrupulous commercial breeders (often called “puppy farms”) in the UK, and a rise in the importation of dogs, including illegal “puppy smuggling” from overseas [9,10]. The practices associated with the breeding and supply of dogs are closely tied to the health and welfare of these animals. Intensive breeding through puppy farming, including those illegally imported into the UK, has negative impacts on dog health and behaviours [10,11]. Increased prospective owner awareness of the implications of sourcing on dog welfare may improve the acquisition decisions made and benefit canine welfare. It has also been the case that while people continue to acquire dogs, many dogs have been relinquished every year [12,13]. There are numerous reasons for this, including owners not having access to information about dogs’ needs or understanding the time, effort, and costs involved in dog ownership [14,15,16,17]. Access to this information, through pre-acquisition research, may improve owner expectations and reduce the risk of relinquishment [17]. In the UK, various online resources have been developed by charities and professional organisations that are designed to provide information for prospective owners about responsibly acquiring dogs—particularly puppies—and dog ownership, e.g., [18,19,20,21]. However, the extent to which these resources are accessed or influence subsequent buying behaviour is not known [22]. There is an urgent need to better understand acquisition decisions so that appropriate strategies can be developed to enable prospective owners to make informed decisions and help safeguard dog welfare.

There is currently limited knowledge about how prospective dog owners decide how and where to acquire their dogs. In particular, we know little about whether prospective owners undertake preparatory research. Previous research suggests that many prospective dog owners undertake some sort of research prior to acquiring a dog. The majority (84.3%) of dog owners who responded to a 2015 survey conducted by Packer et al. [23] stated that they had undertaken pre-acquisition research. Similarly, Kuhl et al.’s 2013 study [24] found that over three-quarters (78.9%) of owners reported looking for information prior to acquiring their dogs. Kuhl (2021) suggested that some owners may consider themselves to have adequate experience so as not to require further research [25]. A study into so-called “Pandemic Puppies” also investigated pre-purchase behaviours of UK owners who acquired their dog as a puppy (aged under 16 weeks) during 2019 or the 2020 COVID-19 pandemic [2]. In this population, almost half (46.7%) of owners who acquired their puppy in 2019 completed pre-purchase research but half (50.3%) did not because they considered themselves to be experienced dog owners. More owners who purchased a puppy in 2020 undertook research (58.1%); however, the difference between years was explained by ownership experience, as for the latter group, only 38.9% considered themselves experienced enough to have not needed to undertake pre-purchase research. Just 3% of owners across both year groups had both not completed any pre-purchase research and considered themselves to be inexperienced owners [2]. Kuhl et al. [24] found that owners of pedigree dogs were more likely than owners of non-pedigree dogs or a mix of pedigree and non-pedigree dogs to have sourced information prior to acquiring their dog (85.6% versus 71.9%). Burnett et al. (2022) found that pre-purchase research was more common amongst owners of designer crossbreeds (73.0%) compared to purebred puppies (48.6%); however, a higher proportion of purebred owners considered themselves to already be experienced dog owners [26].

This study aimed to better understand the preparatory research undertaken by current and potential dog owners in the UK. Specifically, we were interested in understanding, prior to acquiring a dog: (1) do prospective owners look for information or for advice; and (2) what factors influence whether prospective owners undertake research? This study is the first that we are aware of to investigate a number of factors such as owner and dog demographics in relation to pre-acquisition information gathering. As such, this adds novel insights to the extant research into this area.

## 2. Materials and Methods

This study used a convergent mixed methods design, with data collected through a survey and interviews, as shown in Figure 1 after Ref. [27]. Data were collected largely in parallel, analysed independently, and interpreted together in a comparative and contrasting way [28]. Data were collected as part of a wider study investigating various aspects of dog acquisition [27,29].

### 2.1. Ethics Statement

Ethical approval for this study was granted by the Dogs Trust Ethical Review Board (reference numbers: ERB018 and ERB019). All participants were provided with an informed consent statement prior to participation in the study. This outlined the purpose of the study, described how data—especially any personal data—would be stored and used, explained that participation was voluntary, and provided instructions on how to withdraw from the study. Informed consent was obtained on the first page of the survey from survey participants, by signature for participants who were interviewed face-to-face, or verbally (audio recorded) for participants who were interviewed remotely. Participants were required to be aged 18 years or over and living in the UK. No payment or incentives were offered to participants.

### 2.2. Data Collection

#### 2.2.1. Survey Design and Content

A self-completion online survey was designed to collect predominantly quantitative data about the experiences of current and potential dog owners. Qualitative data were also collected through free text responses. Questions were developed by the authors and were informed by a review of the current literature [30]. The survey was hosted on the online survey platform SmartSurvey^TM^ (https://www.smartsurvey.co.uk/ (accessed on 4 October 2021)). Prior to the launch, the survey was piloted twice: firstly, with 12 members of Dogs Trust staff who were not involved in developing the survey and, secondly, with 110 current or potential dog owners, who were recruited through two promotional posts on the Facebook page of “Generation Pup” (Generation Pup (https://generationpup.ac.uk/ (accessed on 2 May 2019)) is an ongoing longitudinal cohort study of dogs. Generation Pup has ethical approval from the University of Bristol Animal Welfare Ethical Research Board (UIN/18/052), the Social Science Ethical Review Board at the Royal Veterinary College (URN SR2017-1116), and the Dogs Trust Ethical Review Board (ERB009). Survey links were posted on the Generation Pup Facebook page (https://www.facebook.com/generationpup/) (accessed on 29 April 2019))). Following these pilots, minor changes were made to the logic of the survey to make respondents’ progression easier.

The survey took approximately 20 min to complete and asked a variety of questions related to pre-acquisition research as part of a wider study on dog acquisition. It also included questions about the demographics of owners and (where applicable) their dogs and asked whether participants would be willing to be contacted about further research opportunities. Participants were asked if they currently owned at least one dog (thus were a “current” owner) and if they were considering acquiring a/another dog(s) in the future (“potential” owners). Owners who owned more than one dog were asked to provide responses for the dog they had acquired most recently. If more than one dog was acquired at the same time, respondents were asked to answer for the dog whose name was first alphabetically. For the purposes of this study, current owners who were considering getting another dog were only asked questions retrospectively about any research they undertook prior to acquiring their current dog: they were not asked questions about whether they were currently looking (or planning to look) for information or advice with regards to their future dog(s). All relevant survey questions and response options can be found in Appendix A.

Current owners (n = 8050) were asked the question “Did you look for any information or ask anyone for advice before getting your dog?”. Those respondents who selected “No” were not asked any further questions about pre-acquisition research. Those respondents who selected “Yes” were asked further about their pre-acquisition research, including “What information or advice did you look for before getting your dog?” (open-ended question). Although details of the nature of this information or advice are not reported in this article (but are the focus of a forthcoming paper), some respondents included information within this survey item about why they undertook research; thus, relevant responses have been included in the analysis for this study. All current owners (regardless of whether they had undertaken any research) were also asked the optional open-ended question “What advice would you give to potential dog owners about buying or rehoming a dog?”.

Potential owners (n = 2884) were asked similar questions to those outlined above for current owners. The question “Have you looked for any information or asked anyone for advice about getting a dog?” could be answered in four ways: “Yes”, “No but I plan to”, “No and I don’t plan to”, or “I haven’t thought about this yet”.

#### 2.2.2. Survey: Participant Recruitment

The survey was live for three months at the end of 2019 (25th September 2019–31st December 2019). It was promoted predominantly through Dogs Trust via social media posts, correspondence with supporters (e.g., e-newsletters and magazine), the Dogs Trust Contact Centre and rehoming centres. Thus, the sample was a convenience sample; however, some promotions were targeted towards males and those who were not supporters of Dogs Trust, through a paid Facebook advertisement. This was to improve reach and to increase representation of these groups, given that male participants are underrepresented in studies of human–animal relationships [31]. Further information on how participants were recruited to the survey can be found in Appendix A.

#### 2.2.3. Interviews

Interviews were conducted with current and potential dog owners to gain a deeper understanding of aspects of the dog acquisition process. We conducted two types of interviews as part of the study: pre-arranged and ad hoc. Each interview was conducted by one of three authors (R.M., K.E.H., or R.M.C). All interviews explored owners’ experiences of dog acquisition, including whether they had conducted any research prior to acquiring their current or prospective dog. Both types of interviews followed a semi-structured guide. This was piloted in a similar manner to the survey: firstly with 12 members of Dogs Trust staff; secondly with 5 respondents to the pilot survey. The interview guide was not changed following pilot interviews, thus data from pilot interviews were included in the analysis for the overall study. Interview guides can be found in Appendix A.

Pre-arranged interviews were used to stimulate in-depth discussion about the dog acquisition process and were conducted between April 2019 and March 2020. They were conducted with current (n = 24) and potential (n = 8) dog owners. Of the 24 current owners, 3 were considering acquiring another dog in the near future. Interviewees were recruited through the survey (n = 15), pilot survey (n = 5), or were members of Dogs Trust staff (n = 12). Interviews were conducted remotely via telephone (n = 22) or face-to-face (n = 10). The majority of interviews were with individuals (n = 25), but 7 individuals were involved in group interviews with 2 or 3 participants in each (these were all with Dogs Trust staff and were trialled as part of the pilot: it was decided to focus on individual interviews following this, mainly for logistical purposes). All interviewees participated once, with the exception of one participant who also was involved in two additional follow-up interviews following acquisition of their dog. Interviews lasted between 17 and 60 min in length (mean = 33 min, median = 29 min). With participants’ consent, interviews were audio recorded and later transcribed *intelligent verbatim*, (i.e., false starts, pauses, laughter, and filler words such as “um” and “err” were omitted).

Ad hoc interviews were undertaken to collect data from a broader range of dog owners than were likely to be reached through the pre-arranged interviews. These were conducted at 23 “Responsible Dog Ownership” events across the UK, organised by Dogs Trust Regional Campaigns staff, between May and December 2019. At these events, dog owners could gain advice on diet, exercise, and enrichment, and (excluding Northern Ireland events) veterinary nurses provided free basic health checks and microchipping. The locations for these events were determined using findings from Dogs Trust Stray Dog Survey data, e.g., [32], and discussions with community partners (e.g., dog wardens and housing association staff) about local hotspots for dog-related issues and areas of deprivation. Thus, these events were sometimes held in association with local authorities or housing associations and often took place in parks or community centres.

For ad hoc interviews, event attendees were asked whether they would be happy to be interviewed, either while they were waiting to be seen by a member of staff, or after they (and their dog(s)) had been attended to. If they were willing to participate, they were interviewed on-the-spot. In total, 142 current dog owners or carers (or sets of owners, where a dog was accompanied by more than one person) and 2 potential owners were interviewed. With consent, 44.4% of these interviews were audio recorded. Recorded interviews lasted between 2 and 26 min in length (mean = 11 min, median = 11 min) and were transcribed as per the pre-arranged interviews. Where interviews were not recorded (either due to participants not giving consent or where events were thought to be too noisy to enable clear audio recordings), the researcher made handwritten notes during and immediately after interviews, which were subsequently digitised by the researcher who conducted the interview.

### 2.3. Data Analysis

#### 2.3.1. Quantitative Data Analysis

Initial data cleaning was completed in Microsoft Excel and IBM SPSS (v. 26). Responses to relevant closed-ended survey questions were summarised with descriptive statistics using IBM SPSS (v. 26) and R v. 4.1.2; [33]. Univariable and multivariable analyses were also completed using R (v. 4.1.2). These were used to compare responses given by different groups of respondents and to determine the relative contributions and relationships between variables.

#### 2.3.2. Qualitative Data Analysis

The aims of the qualitative analysis were to identify further factors that may influence dog owners’ decisions to seek information or advice before acquiring their dog and explore the range of dog owners’ experiences of conducting such research. Interview transcripts and relevant free-text survey responses were imported into NVivo (v. 12). These data were then analysed using a thematic analysis approach [34] by two authors (R.M. and K.E.H.). The process of thematic analysis began with familiarisation of the data by reading through a selection of free-text survey responses and interview transcripts. Initial coding of data was inductive, meaning that the coding was driven by the content of the responses, rather than using pre-determined codes. Codes were then grouped into categories and then into potential themes that linked the concepts within categories and represented overall patterns in the data.

All interview data were initially coded and collated into potential themes by one author (K.E.H) as part of the wider acquisition study. Subsequently, interview data extracts relevant to the current study (i.e., data related to pre-acquisition information seeking) were independently coded and grouped into potential themes by R.M., prior to collaborative review of coding and theme development (by R.M. and K.E.H.).

Free-text survey data relevant to this study were collected across four open-ended questions. During initial coding, the researchers performed an ongoing appraisal of the “information power” [35] of the responses to each question. Information power refers to the amount of information contained within a sample, relevant to the study, and suggests that fewer participants are required in samples with more information power [35]. For two of the questions, the researchers determined that a subsample of responses had sufficient information relevant to answer the question and met the aim of this study’s qualitative analysis, as outlined above. Once it was recognised that a subsample likely offered ample information power, and that incorporating additional responses was unlikely to elicit new codes or generate new understanding for codes [36], a quasi-random sampling approach was applied to the remaining data. This involved coding every 25th additional response. Further detail on how qualitative survey data were coded is presented in Appendix A.

Where direct quotes are included within this study, names have been omitted to protect participant confidentiality. Instead, a unique identifier and brief description of the participant (dog ownership status and mode of data collection) are given.

## 3. Results

### 3.1. Survey Results

The survey was started 15,350 times. Following data cleaning and deduplication, 11,265 of these responses were suitable for analysis. These comprised 8381 current owners and 2884 potential owners; however, of these current owners, 115 had bred their own dogs and 216 were not involved in the decision to acquire a dog. Therefore, the initial sample sizes reported here are for 8050 current and 2884 potential owners.

#### 3.1.1. Participant Demographics

The majority of survey participants were female (88.3% of current and 79.9% of potential owners). Respondents represented age groupings from 18 to 85 years and above, with 45–54 years being the most common age category for current owners (23.8%) and 55–64 years being the most common for potential owners (20.8%). Respondents resided in all four UK nations, but the majority were based in England (84.2% of current and 82.4% of potential owners). Additional information on participant demographics can be found in Appendix A.

#### 3.1.2. Dog Demographics

The majority of current owners (62.8%) had acquired their dog within 5 years prior to the completion of the survey. Over half of the dogs (54.4%) were acquired as puppies of 6 months or younger. Just over half (54.9%) were a specific breed (e.g., Labrador Retriever) and 22.9% were a mix of two specific breeds (e.g., Cockerpoo/Cocker Spaniel × Poodle cross). Most dogs were acquired from a charity or rehoming centre (43.6%) or from a breeder (39.6%). Further information on dog demographics can be found in Appendix A.

### 3.2. Do Prospective Owners Look for Information or Advice before Acquiring a Dog?

Of the current owners surveyed, just over half stated that they had looked for advice or information prior to acquiring their dog (54.4%, 95% CI [53.3%, 55.5]). Two-thirds of potential owners reported already having looked for information (67.8%, 95% CI [66.1%, 69.5%]) and a further 13.7% (95% CI [12.5%, 15.0%]) were planning to undertake research. Potential owners were significantly more likely to report having undertaken research than current owners, *X*^2^ (1, *N* = 10,934) = 155.1, *p* < 0.001.

### 3.3. What Factors Influence Whether Prospective Owners Undertake Research Prior to Acquiring a Dog?

Multivariable analysis using survey responses suggested numerous factors that contribute to the likelihood of research being undertaken, for current (Table 1) and potential (Table 2) owners.

Previous dog ownership appeared to be an important factor: those who had lived with a dog or dogs as a child and an adult, and thus may be considered to have the most previous experience of ownership, were the least likely to undertake research (44.8% of current owners and 77.4% of potential owners). Compared to those who had owned dogs as an adult and a child, those who had previously lived with a dog or dogs as an adult (only) were 1.2 times more likely for current owners, and 1.4 times more likely for potential owners, to conduct research. Those who had (only) lived with a dog or dogs as a child were 2.5 times (current) and 3.6 times (prospective) more likely to conduct research. Those who had never lived with a dog previously were the most likely to have undertaken research (4.6 times and 11.1 times, respectively).

Among current owners, this pattern appeared to be mirrored to some extent based on ownership of the same breed or type of dog, with those current owners who had not previously owned the same breed or type of dog as their current one being significantly more likely to have undertaken research. However, this variable was not included simultaneously in the multivariable analysis with the variable “previous dog ownership” due to issues of collinearity. Further information can be found in Appendix A.

**Table 1 animals-13-01015-t001:** Multivariable analysis of factors that affect whether research was undertaken prior to acquiring a dog for current owners (multivariable analysis n = 7279; univariable analyses n are given for each variable).

	Undertook Research				Univariable Analysis					Multivariable Analysis (n = 7279)				
Variable	Yes	Total	%	95% CI	Odds Ratio ^1^	2.50%	97.50%	z Val.	*p* Val.	Odds Ratio	2.50%	97.50%	z Val.	*p* Val. ^2^
Previous ownership (n = 8050)														
Previously lived with a dog/dogs as an adult and as a child	1588	3546	44.78%	43.15%, 46.42%	Ref				<0.0001	Ref				<0.0001
Previously lived with a dog/dogs as an adult	1236	2458	50.28%	48.31%, 52.26%	1.25	1.13	1.38	4.20	<0.0001	1.22	1.09	1.36	3.39	0.0007
Previously lived with a dog/dogs as a child	850	1188	71.55%	68.92%, 74.04%	3.10	2.69	3.58	15.58	<0.0001	2.53	2.16	2.96	11.60	<0.0001
First time lived with a dog	707	858	82.40%	79.71%, 84.81%	5.77	4.79	6.97	18.30	<0.0001	4.61	3.75	5.66	14.56	<0.0001
Age of owner (n = 7987)														
75 years or older	65	196	33.16%	26.94%, 40.03%	Ref				<0.0001	Ref				<0.0001
65–74 years	463	1126	41.12%	38.28%, 44.02%	1.41	1.02	1.94	2.09	0.0364	1.31	0.92	1.87	1.48	0.1388
55–64 years	904	1821	49.64%	47.35%, 51.94%	1.99	1.46	2.71	4.32	<0.0001	1.55	1.10	2.20	2.49	0.0128
45–54 years	1036	1916	54.07%	51.83%, 56.29%	2.37	1.74	3.24	5.45	<0.0001	1.59	1.12	2.25	2.60	0.0092
35–44 years	731	1206	60.61%	57.83%, 63.33%	3.10	2.25	4.27	6.95	<0.0001	1.76	1.23	2.52	3.08	0.0021
18–24 years	311	474	65.61%	61.22%, 69.75%	3.85	2.70	5.47	7.49	<0.0001	2.02	1.35	3.03	3.43	0.0006
25–34 years	839	1248	67.23%	64.57%, 69.78%	4.13	3.00	5.69	8.69	<0.0001	2.28	1.58	3.28	4.43	<0.0001
Work with dogs (n = 7850)														
Currently work with dogs	386	776	49.74%	46.23%, 53.25%	Ref				<0.0001	Ref				<0.0001
Previously worked with dogs	380	786	48.35%	44.87%, 51.84%	0.95	0.78	1.15	-0.55	0.5810	1.23	0.99	1.53	1.83	0.0667
Never worked with dogs	3409	6116	55.74%	54.49%, 56.98%	1.27	1.10	1.48	3.18	0.0015	1.48	1.25	1.75	4.52	<0.0001
Highest level of education (n = 7373)														
No formal qualifications	107	321	33.33%	28.40%, 38.66%	Ref				<0.0001	Ref				<0.0001
GCSE/National 5 or equivalent	656	1384	47.40%	44.78%, 50.03%	1.80	1.40	2.33	4.53	<0.0001	1.42	1.08	1.86	2.50	0.0123
A level/Scottish Higher or equivalent	525	965	54.40%	51.25%, 57.52%	2.39	1.83	3.11	6.45	<0.0001	1.51	1.13	2.01	2.81	0.0049
Foundation degree/Higher National Diploma (HND) or equivalent	570	1085	52.53%	49.56%, 55.49%	2.21	1.71	2.87	5.97	<0.0001	1.56	1.18	2.06	3.10	0.0019
University degree (e.g., BA, BSc) or equivalent	1362	2305	59.09%	57.07%, 61.08%	2.89	2.26	3.70	8.44	<0.0001	1.83	1.40	2.38	4.44	<0.0001
Postgraduate degree (e.g., MA, MBA, MSc, PhD) or equivalent	831	1313	63.29%	60.65%, 65.86%	3.45	2.66	4.46	9.41	<0.0001	2.27	1.72	3.00	5.76	<0.0001
Source of dog (n = 8050)														
Friends or family/community	393	979	40.14%	37.12%, 43.25%	Ref				<0.0001	Ref				<0.0001
Private/third party seller	208	454	45.81%	41.29%, 50.41%	1.26	1.01	1.58	2.02	0.0431	1.17	0.91	1.50	1.20	0.2299
Charity/rehoming centre	1655	3427	48.29%	46.62%, 49.97%	1.39	1.21	1.61	4.50	<0.0001	1.52	1.28	1.80	4.74	<0.0001
A dog breeder	2125	3190	66.61%	64.96%, 68.23%	2.98	2.57	3.45	14.49	<0.0001	2.38	1.99	2.83	9.61	<0.0001
Breed or type of dog (n = 7596)														
Mix of breeds or types	844	979	45.82%	43.56%, 48.10%	Ref				<0.0001	Ref				<0.0001
Mix of two specific breeds	1012	3190	56.60%	54.29%, 58.88%	1.54	1.35	1.76	6.48	<0.0001	1.18	1.01	1.38	2.06	0.0395
Specific breed	2525	3427	57.13%	55.66%, 58.58%	1.58	1.41	1.76	8.15	<0.0001	1.37	1.19	1.57	4.43	<0.0001
Year acquired					1.05	1.04	1.07	8.53	<0.0001	1.06	1.04	1.07	7.83	<0.0001
Age of dog when acquired ^3^ (n = 8050)														
Senior adult (7 to <12 years)	195	465	41.94%	37.53%, 46.47%	Ref				<0.0001	Ref				0.0004
Juvenile (>6 months to <1 year)	30	65	46.15%	34.59%, 58.15%	1.13	0.88	1.46	0.98	0.3295	1.11	0.84	1.47	0.72	0.4713
Geriatric (12+ years)	334	755	44.24%	40.73%, 47.80%	1.19	0.70	2.00	0.64	0.5196	1.21	0.67	2.18	0.64	0.5224
Young adult (1 to <2 years)	394	888	44.37%	41.13%, 47.65%	1.07	0.85	1.34	0.55	0.5818	1.23	0.95	1.58	1.58	0.1137
Mature adult (2 to <7 years	755	1496	50.47%	47.94%, 53.00%	1.38	1.12	1.70	3.04	0.0024	1.41	1.12	1.77	2.94	0.0033
Puppy (0–6 months)	2673	4381	61.01%	59.56%, 62.45%	2.17	1.78	2.63	7.81	<0.0001	1.59	1.25	2.03	3.80	0.0001

^1^ Ordering is based on multivariable model, hence some Odds Ratios < 1; ^2^ The *p* values given for each reference row are likelihood ratio test *p* values; ^3^ Age categorisations based on [37].

**Table 2 animals-13-01015-t002:** Multivariable analysis of factors that affect whether research was undertaken prior to acquiring a dog for potential owners (multivariable analysis n = 2272; univariable analyses n are given for each variable).

	Undertook Research				Univariable Analysis					Multivariable Analysis (n = 2272)				
Variable	Yes	Total	%	95% CI	Odds Ratio ^1^	2.50%	97.50%	z Val.	*p* Val.	Odds Ratio	2.50%	97.50%	z Val.	*p* Val. ^2^
Previous ownership (n = 2861)														
Previously lived with a dog/dogs as an adult and as a child	1086	1403	77.41%	75.14%, 79.52%	Ref				<0.0001	Ref				<0.0001
Previously lived with a dog/dogs as an adult	689	880	78.30%	75.45%, 80.90%	0.72	0.50	1.02	−1.83	0.0679	1.36	1.06	1.74	2.43	0.0151
Previously lived with a dog/dogs as a child	284	303	93.73%	90.86%, 96.00%	3.15	1.26	7.87	2.45	0.0141	3.56	2.02	6.27	4.39	<0.0001
Never lived with a dog	269	275	97.82%	95.21%, 99.11%	5.09	1.59	16.28	2.74	0.0061	11.10	4.07	30.28	4.70	<0.0001
Age of owner (n = 2865)														
65–74 years	274	374	73.26%	68.55%, 77.50%	Ref				<0.0001	Ref				<0.0001
55–64 years	449	599	74.96%	71.33%, 78.27%	1.30	0.78	2.18	1.01	0.3120	1.05	0.74	1.49	0.26	0.7932
75 years or older	60	88	68.18%	57.84%, 77.01%	0.71	0.32	1.56	−0.85	0.3960	1.09	0.59	1.99	0.27	0.7870
45–54 years	457	573	79.76%	76.27%, 82.85%	1.17	0.70	1.95	0.61	0.5406	1.45	1.00	2.10	1.95	0.0515
35–44 years	388	453	85.65%	82.11%, 88.59%	3.58	1.72	7.48	3.40	0.0007	1.60	1.07	2.40	2.29	0.0218
25–34 years	507	562	90.21%	87.46%, 92.42%	2.76	1.47	5.18	3.17	0.0015	2.95	1.90	4.57	4.83	<0.0001
18–24 years	198	216	91.67%	87.14%, 94.73%	5.75	1.73	19.12	2.85	0.0044	3.34	1.76	6.36	3.68	0.0002
Work with dogs (n = 2716)														
Currently work with dogs	275	361	76.20%	71.5%, 80.3%	Ref				0.0113	Ref				0.0124
Previously worked with dogs	133	155	85.80%	79.4%, 90.5%	0.12	0.02	0.93	−2.03	0.0428	1.04	0.59	1.83	0.13	0.8948
Never worked with dogs	1813	2200	82.40%	80.8%, 83.9%	0.13	0.02	0.92	−2.04	0.0413	1.58	0.94	2.64	1.74	0.0817

^1^ Ordering is based on multivariable model, hence some Odds Ratios < 1; ^2^ The *p* values given for each reference row are likelihood ratio test *p* values.

Whether someone currently or had previously worked with dogs was significantly associated with whether research was undertaken. Current and potential owners who had experience of working with dogs (at the time of, or previous to, survey completion) were less likely to have undertaken research than those who had never worked with dogs. Current owners who had never worked with dogs were 1.5 times more likely to have undertaken research than those who worked with dogs at the time of survey completion.

The age of owner was a significant factor, with younger prospective owners being more likely to undertake research prior to acquiring a dog: just a third of current owners aged 75 years or older had undertaken any research prior to acquiring their most recent dog compared to approximately two-thirds of 25–34 (67.2%) and 18–24 year olds (65.6%). Current owners aged 25–34 years old were 2.3 times more likely to have undertaken research prior to acquiring their most recent dog, compared to those aged 75 years or older, and similar patterns were seen among potential owners.

Among current owners, those with formal education qualifications were also significantly more likely to have undertaken research prior to acquiring a dog. A third of those who had no formal qualifications had undertaken research. The odds of undertaking research increased with increasing levels of formal education, with those with a postgraduate qualification having 2.3 times greater odds of having undertaken research (63.3%). These differences were not significant among potential owners when included in the multivariable analysis. However, the pattern of effect was similar when education was considered in isolation, with research more likely among people with formal education (Appendix A).

A number of factors related to the dog acquired also appeared important in terms of whether any pre-acquisition research was undertaken by current owners. Specifically, the odds of undertaking research were higher when prospective owners went on to acquire their dog from a breeder, when a dog was a specific breed or a mix of two specific breeds, when a dog was acquired as a puppy, and when a dog was acquired more recently. If a causal relationship exists with these variables, the direction of the effect is unclear, nor whether these associations could be due to the confounding effect of other, unmeasured, variables. Further information, including univariable results relevant to gender, having children under 18 living at home, employment status, and whether research was undertaken, can be found in Appendix A.

Qualitative data offer additional suggestions as to why prospective owners choose to undertake research prior to acquiring a dog. Some owners commented on how important it was for them to have looked for information prior to acquiring a dog, and many recommended that all potential owners should “*do your research*”. For some prospective owners, searching for information was an important part of deciding whether to acquire a dog:


*“Whether I would have the time and resources needed to give a dog a good home.”*
(Current owner, survey ID 3136)

Others discussed how they were motivated to undertake research as they had their potential dogs’ interests in mind. Several commented on how they sought information to ensure that they could offer a good home and appropriate lifestyle that would ensure a dog’s wellbeing. Thus, undertaking research appeared important in preparing for the arrival of a dog:


*“Everything I may need to know to look after him/her to the best of my ability. And make sure she has everything to fit her needs.”*
(Potential owner, survey ID P3287)

Furthermore, some prospective owners appeared to recognise the value of completing research in preparation for managing the emotional drivers associated with acquisition:


*“Do your homework as your heart takes over when you go looking.”*
(Current owner, survey ID 1274)

As identified with quantitative analysis, previous ownership and experiences with dogs appeared to be an important factor when undertaking research. Those with little previous experience, including of a breed or life stage, sometimes commented on this as being a motivator when choosing to undertake research:


*“As she’s our first dog, we did a lot of research into different breeds and their personalities.”*
(Current owner, survey ID 1549)


*“We haven’t had a puppy between us before […] so we had a lot of studying to do about puppies.”*
(Current owner, survey ID 1828)

A number of prospective owners who had previously owned dogs described how they conducted research to update their knowledge prior to looking for their next dog:


*“I have previously owned 3 dogs and have researched each one. Over the last month, I’ve been giving myself a refresher course.”*
(Potential owner, survey ID 2841)

In contrast, some who had previously owned dogs noted that they only needed to find information related to specific areas that they were less familiar with, often highlighting that they thought they knew about other aspects of dog ownership and thus did not need any more information on these:


*“Quite knowledgeable on dogs in general, so was more specific to specific dog and situation and breed that I’ve never had before.”*
(Current owner, survey ID 1900)

Sometimes the need to undertake research appeared to be linked to the amount of time since a prospective owner had last had a dog, with those without recent experience suggesting that they needed to update their knowledge or find current advice:


*“As I had not owned a dog for 20 years.”*
(Current owner, survey ID 8340)

Some prospective owners noted that, regardless of previous experience of similar breeds or circumstances, they were always keen to learn more:


*“We did have experience of this [rehoming a Greyhound], having rehomed two retired racers, but the more advice the better!”*
(Current owner, survey ID 8292)

Despite the value many prospective owners placed on seeking information or advice, a considerable proportion of prospective owners did not conduct any research prior to acquiring a dog. In interviews, a number of barriers became apparent. Some suggested that due to their previous experiences as a dog owner they felt that they did not need any information or advice. For example, one interviewee described how they did not conduct any research before acquiring their current dog as they had owned dogs before and felt they knew everything they needed to:


*“I didn’t [do any research] because I’m one of these people that thinks they know everything; do you know what I mean? Because I had had animals and had dogs and done dog training and dog trials, I had a very high opinion of myself. What, you know dog temperaments and how to train them and all that kind of thing, so no I didn’t get any advice whatsoever.”*
(Current owner, interview ID B1RM1201)

Previous experiences of a chosen breed seemed a particularly important barrier against undertaking any research:


*“We didn’t [do any research] as we’d had that breed before.”*
(Current owner, survey ID 2279)

## 4. Discussion

Undertaking research prior to acquiring a dog is thought to be important for a successful dog–owner bond and the dog’s future wellbeing [14,15,16,17]. Despite this, few studies have investigated the links between pre-acquisition research, acquisition behaviours, and ownership. This study used mixed methods to understand factors influencing pre-acquisition research.

This study is in keeping with the findings of previous studies, e.g., [23,24,38] in that the majority of prospective dog owners undertook some research prior to acquiring a dog, although we found that this was less common than in these previous studies. Our study found that those aspiring to acquire a dog at the time of survey completion were more likely to have undertaken pre-acquisition research (or planned to complete research) than those who already owned a dog did prior to acquiring their current dog. These differences might be due to multiple factors, including recall bias among current owners, recent shifts in behaviour towards undertaking pre-purchase research, or the source that potential owners planned to use (a far higher proportion of potential than current owners found out about our survey through Dogs Trust when enquiring about rehoming a dog—see Appendix A, Table 1). It is also possible that potential owners who had not yet completed any research, but planned to, were over-optimistic in their research aspirations.

The factors that influence pre-acquisition research are varied. Among those who did not undertake research, previous experience with dogs was an important factor: those with more dog ownership experience were less likely to undertake research, as in previous studies [2,25]. Those working with dogs at the time of the survey completion were also less likely to have undertaken any research. Although self-described “experience” appears important, we do not know what this experience entailed for our respondents. For example, the number of dogs, length of dog ownership, or extent of dog caring responsibility are unknown. The relative success of previous dog ownership experiences in terms of dog wellbeing and strength of human–animal bonds are also unknown [39]. Equally, we do not know any details about the nature or period of working with dogs, or—for current owners—whether they had worked with dogs at the time they acquired their most recent dog. Regardless, qualitative data confirm that perceived views of their own experience with dogs was an important factor for prospective owners and is worthy of consideration by those involved in designing resources and interventions to influence decisions related to dog acquisition.

The age of the owner was an important factor, with younger prospective owners being more likely to have sought information or advice. This might suggest a greater importance placed by younger people on research, or that younger people are more easily able to access resources due to greater internet use and digital literacy [40,41,42]. Younger age groups will also likely have less dog ownership experience. Education was also important, with those current owners who attained the highest level of formal study being the most likely to have undertaken research. This may be indicative of an awareness of the importance of research or the ability to access resources. Regardless, these findings highlight that interventions to increase research and preparation prior to dog acquisition should reach across age and demographic groups.

Among current owners, the likelihood of conducting research was associated with the source of their current dog. Those who acquired their dog from breeders were most likely to have undertaken research, followed by those who acquired from rehoming centres. It is not clear whether research leads prospective owners to a particular source or whether those who intend to use a particular source (i.e., a breeder) are more motivated to undertake research. This may be influenced by the degree of financial investment with different sources. Those who had acquired their dog from friends, family, or the community may be more likely to be unplanned or less planned acquisitions, potentially driven by emotions more than intention [29], and may not allow time for pre-acquisition research. The type of dog acquired was also a motivator to pre-acquisition research: similar to previous research by Kuhl et al. [24], those who acquired specific breeds or mixes of two specific breeds appeared more likely to undertake research than those who acquired a dog of mixed breeds or unknown type. This may be confounded with source and possibly represents similar motivations for undertaking research. People tended to be more likely to undertake research if they went on to acquire a puppy. Although the causality of this relationship is unknown, qualitative research suggested that life stage was a driver for research amongst prospective owners. Those who acquired their dog more recently were more likely to have undertaken research. This might be indicative of increased awareness of the value of conducting research and greater visibility of, or easier access to, resources. Alternatively, recall bias may have affected responses from those who acquired their dogs longer ago.

There are likely other motivators and barriers to pre-acquisition research, and it should be noted that no questions were asked about why participants did *not* search for information or advice. It is possible that those who chose not to undertake research were unaware of the benefits of research (as suggested by [24]). Equally, prospective owners may have been restricted by practicalities: they may have acquired a dog within a time frame that did not allow for preparatory research, for reasons including unplanned acquisitions [29].

This study provides insights into the motivators and barriers to undertaking pre-acquisition research that may be useful for developing advice related to dog acquisition. Some experienced owners may be less receptive to messaging as they are more likely to rely on their existing knowledge and believe further research to be unnecessary. Successful messaging directed towards those who have previously owned dogs will need to overcome this barrier, perhaps by focusing on the importance of updating knowledge or targeting particular gaps in knowledge. Less experienced owners are more likely to undertake research, so reaching them may involve providing readily available resources in a format that are accessible to them. Given that almost a fifth of first-time dog owners did not recall undertaking any research before acquiring their current dog, there is clearly an opportunity to better understand how to reach more prospective first-time owners. Our research also demonstrates how demographics may affect the likelihood of undertaking pre-acquisition research. Providing resources which are relevant to different demographic groups, through different media channels and styles, may increase accessibility.

### 4.1. Strengths and Limitations

This is the UK’s largest study of pre-acquisition research among prospective dog owners that we are aware of and adds to the existing evidence base. The mixed methods approach allowed the integration of quantitative and qualitative data, enabling a comparison of findings and greater credibility. The collection of free text survey items allowed for a wide breadth of responses to be documented, whereas semi-structured interviews yielded deeper insights into pre-acquisition behaviours. Collecting data from both current and potential owners enabled understandings from retrospective and current perspectives to be observed. Follow-up surveys will add further insight into this area.

This study has several limitations. The methods involved using a convenience sample, with a bias towards Dogs Trust supporters. This was particularly prominent among potential dog owners, of whom a large proportion were invited to participate in the survey on enquiring to Dogs Trust about adopting a dog. Respondents were self-selecting and there was a bias towards female respondents. Although this is common in dog-related surveys [37], the underrepresentation of males and those who prefer to self-identify means that further research is needed. Caution is needed if attempting to generalise findings to other populations. All survey responses were self-reported; thus, it is not possible to validate findings. Current owners completed the survey retrospectively. Although the majority (62.8%) of current owners had acquired the dog that they completed the survey about within the 5 years prior to survey completion, there is the potential for recall bias. Indeed, a small number of owners had acquired their dogs as long as 19 years prior to survey completion. Certain demographic information was reported at the time of data collection and may not have been the same as at the point of dog acquisition (e.g., owner age category or highest level of education). A cautionary note should be applied to the analysis of data related to potential owners: while these data can be used for hypothesis generating, further research is warranted.

### 4.2. Future Work

This study reports primarily on prospective owners who undertook research. Although our analyses reveal interesting insights into the factors which may act as motivators or barriers to research, we did not ask questions about why people did not undertake research, either as specific survey items, or in interviews. Inclusion of this insight is worthy of consideration in future studies. This study has not considered how ownership of another dog at the time of acquiring a new dog influences whether research is undertaken. Although we found that ownership experience is an important barrier to undertaking research, we do not know if or how ownership of multiple dogs may influence this. We also have limited understanding of unplanned acquisition and how this relates to pre-acquisition research. Although we know that many prospective owners undertake research, we cannot comment on the quality of research nor whether the information or advice received was correct. Nor do we know how different facets of pre-acquisition research may impact dog acquisition, future ownership, relinquishment, development of the human–animal bond, and dog welfare. These are complex areas to consider but future research that attempts to account for these could be of considerable value. Additional data collection with a wider demographic reach may allow generalisations across different populations and be of use to those interested in targeted interventions related to pre-acquisition behaviours.

Data reported here are from a wider study into dog acquisition, which includes more detailed investigation of pre-acquisition research. A forthcoming publication will focus on where prospective owners look for information, what information prospective owners search for, how long prospective owners spend on research, and whether prospective owners find all the information they want. We have collected additional data from prospective owners through two follow-up surveys, subsequent to the survey reported here, and hope that these may offer additional insights as to the nature of the research which those potential owners planning to undertake research actually did.

## 5. Conclusions

In conclusion, undertaking research before dog acquisition is important for, and valued by, many prospective owners. However, we found that almost half of (current) owners reported undertaking no research prior to acquiring their most recent dog. Those with previous dog ownership experience are less likely to undertake research than first-time owners and those who are younger or who have achieved a higher level of education are more likely to undertake research. Prospective owners who go on to acquire puppies, dogs of a specific breed, and who source their dog from a breeder are more likely to complete pre-acquisition research. Our findings may be of interest to organisations involved in improving pet welfare, especially those who provide advice related to dog acquisition, and provide insights which could support improved messaging and interventions.

## Figures and Tables

**Figure 1 animals-13-01015-f001:**
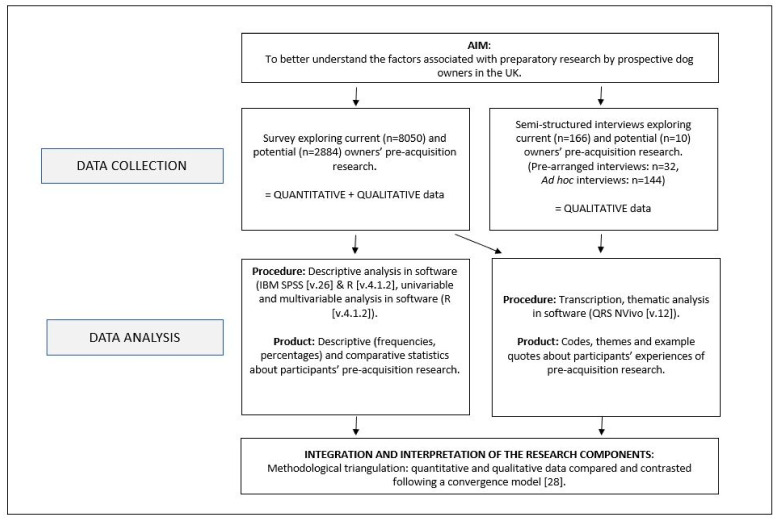
Procedural diagram of the mixed methods study design after Ref. [27].

## Data Availability

The data presented in this study are available on request from the corresponding author. The data are not publicly available due to ethical approval of participant informed consent that included survey respondents being informed that we will remove all personally identifiable information before sharing data with universities and/or research institutions.

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
