# Peer review of "Do Your Homework as Your Heart Takes over When You Go Looking”: Factors Associated with Pre-Acquisition Information-Seeking among Prospective UK Dog Owners"

_animals, 2023, doi:10.3390/ani13061015_

Round 1

Reviewer 1 Report

Thank you for submitting your paper to Animals. Overall it is well written and the study well conducted. I realise that what I say is not within the scope of this study but I would have liked to know:

* if the interviewees had ever surrendered a previous dog to a shelter or pound

*I would also like to know the connection between pre-acquisition research and success of acquisition (in terms of subsequent relinquishment and bond development between adopter and dog).

Could these issues be touched on as of interest and possibly suitable for future research.

Less...

Reviewer 2 Report

The work investigated whether prospective and current dog owners did research prior to dogs acquisition. This topic of research is relevant for dogs' professional in order to improve  services (media, courses, consultancy, events) to better inform future owners and, as a result, improve dogs welfare.

However, the results reported are very general and not informative (see specific comments below). I would suggest the authors to revise the manuscript, highlighting what kind of information the owners search for and through which source and discussing how dog professionals could use their findings to improve their services. 

Introduction

Line 44-45: you may avoid the repetition “in the UK” summarizing the concept of the two sentences in one unique sentence.

Line 54-55: a brief sentence explaining the detrimental effects of “puppy farms” on dogs’ welfare and behavior could help the reader understanding the problem.

Line 90: it would be useful to have a paragraph explaining what is missing about dogs’ pre-acquisition research and which information the authors think is important in order to improve targeted interventions to encourage this activity. 

Data collection 

Line 123: Were the potential dog owners piloting the study person without a current dog or did you included also current owners who were willing to acquire another dog? 

Line 323-325: What is considered as research? -> A person that works with dogs do research all the time she is working (es. training, going to seminars etc..) and this influence her decision on the dog she/he will acquire next. 
In the supplemental material we can see that you asked for which type of research people performed, it would be interesting to have more specific information about that in the results. I think is important for dog professionals to also understand were people source their information about dogs because doing research with bad source is not always a good thing but rather misleading.  

Line 329-330: were younger/older owner more likely to adopt a dog rather than buy it from a breeder (or vice versa)?

Line 383: what about actual dog training courses/consulting previous to acquiring the dog? 

Line 408-411: if this owner actively trained different dogs and performed trials with them, I would consider his/her experience an enormous number of hours “doing homework” and getting knowledge about dogs’ behavior and characteristics. I think that the interviews question about “doing research” is trivial and do not give useful feedback in terms of actual “dog knowledge” of the people interviewed.

This study in focused on a very general “getting knowledge” without actually specifying what this “knowledge and research” is. As a dog trainer and dog behavior researcher I feel that dog professionals and researchers would benefit for more specific information about which specific knowledge people think they need before acquiring a dog (es. breed characteristics/personalities/need vs practical training, behavioral problems, whether they would be willing to start a practical training course). 

Line 441-446: Why didn’t  you investigated the factors cited (number of dogs, length of ownership, extent of dog caring) and I would add “type of working with dogs”, involvement in “dog sports” etc… without this information the abstract concept of “doing research” and “getting knowledge” has no useful practical application (i.e. a previous owner who trained dogs for 20 years has more knowledge about dogs than a future dog owner who did research in internet about breed characteristics).

Line 453-457: The influence of the age could also be due to the fact that older people were the ones who had major experience with dogs (and thus a confounding variable), did you check whether these two factors were correlated?  

Reviewer 3 Report

Good paper with only small comments.

Line 33: a specific breed – is there a breed effect?

Line 156: many dogs have been relinquished – try to quantify (X%)…

Line 213: how many recorded interviews took 2 minutes and why?

Line 283: since puppy-farm seems to be a problem, it might be nice to precise the % of dogs coming from an unknown source.

Line 329 – the age of the owner seems to be a confounding effect as explain in the discussion.
